# Arabic within culture forensic interviews: Arabic native speaking lay-observer truth and lie accuracy, confidence, and verbal cue selection

**Coral J. Dando**[1]*, **Alexandra L. Sandham**[2], **Charlotte Sibbons**[1], **Paul J. Taylor**[3]

**1** Department of Psychology, University of Westminster, London, United Kingdom, **2** The Open University, Milton Keynes, United Kingdom, **3** Department of Psychology, Lancaster University, Lancaster, United Kingdom

* c.dando@westminster.ac.uk

**Data Availability Statement:** All raw data files (SPSS data sheets) is available from the OSF database. The materials and questionnaires are also available on OSF.

## Abstract

Cross cultural differences in behavioral and verbal norms and expectations can undermine credibility, often triggering a lie bias which can result in false convictions. However, current understanding is heavily North American and Western European centric, hence how individuals from non-western cultures infer veracity is not well understood. We report novel research investigating native Arabic speakers' truth and lie judgments having observed a matched native language forensic interview with a mock person of interest. 217 observers viewed a truthful or a deceptive interview and were directed to attend to detailedness as a veracity cue or given no direction. Overall, a truth bias (66% accuracy) emerged, but observers were more accurate (79%) in the truth condition with the truthful interviewee rated as more plausible and more believable than the deceptive interviewee. However, observer accuracy dropped to just 23% when instructed to use the detailedness cue when judging veracity. Verbal veracity cues attended too were constant across veracity conditions with 'corrections' emerging as an important veracity cue. Some results deviate from the findings of research with English speaking western participants in cross- and matched-culture forensic interview contexts, but others are constant. Nonetheless, this research raises questions for research to practice in forensic contexts centred on the robustness of western centric psychological understanding for non-western within culture interviews centred on interview protocols for amplifying veracity cues and the instruction to note detailedness of verbal accounts which significantly hindered Arabic speaker's performance. Findings again highlight the challenges of pancultural assumptions for real-world practices.

## Introduction

Culture can be defined as 'the collective programming' that distinguishes one group from another and the societal markers that determine individual values, attitudes, and behaviours

**Funding:** This work was funded by the U.S. Government's High-Value Detainee Interrogation Group (HIG) awarded to the first author - Contract DJF-16-1200-V-0000737. Statements of fact, opinion and analysis in the article are those of the authors and do not reflect the official policy or position of the HIG or the U.S. Government.

**Competing interests:** The authors have declared that no competing interests exist

[1,2]. Accordingly, individuals from different cultures communicate and behave variously, which can impede understanding of truth and lies since cues to veracity are not pancultural. Yet, current knowledge base, which typically guides practice is largely centred on North American and Western European English-speaking cultures. The forensic implications of focusing on pseudodiagnostic culturally determined behaviours are known to be significant, including wrongful convictions, false confessions and cross-cultural misidentification [3–5]. Towards advancing the sparce literature regarding culturally specific perceptions of truth and lies, we investigate observer veracity judgments of matched culture, matched native language interviews with persons of interest, where interviews are conducted in Modern Standard Arabic (henceforth, Arabic).

Persons of interest is a term used by many law enforcement organisations when referring to individuals suspected of knowing investigation relevant information (IRI), but who at that point in time may not be suspected of criminal activity themselves [6,7]. For example, a person of interest may have witnessed an incident, have some information about a crime that has occurred or is being planned, or might have a close relationship with a person suspected of criminal wrongdoing. Persons of interest are questioned by professionals worldwide using various interview techniques to yield IRI [8–10]. Although not required to make decisions regarding the veracity of accounts provided *per se*, interviewers are expected to be alert to deception. Accordingly, they seek to maximize opportunities to detect truth and lies, gathering information, challenging and/or accepting responses to questions as appropriate, seeking clarification, and deciding when to terminate an interview and when to persist [11–13].

Following an interview with a person of interest, various parties not involved in the interview process itself become concerned with understanding the integrity of accounts provided. Examples include professional non-investigators (e.g., legal and criminal justice employees) and lay-persons (e.g., jury members, assessment panels comprising non-investigative professionals), who observe interviews or portions of interviews with persons of interest [8,14,15]. Understanding lay-persons judgement behaviours is vital since they can be involved in decisions centered on whether to further investigate, the likelihood of a successful prosecution, judging guilt or innocence, and whether responses to cross examination during court proceedings correspond with accounts provided in the interview, for example.

Lay-persons have no investigative experience nor expertise, they simply observe and listen. Nonetheless, lay-observers are an important element of many criminal and civil investigations and formal proceedings worldwide [16–18]. The psychological deception literature concerning lay-observer judgements of truth and lies is generally consistent. Lay-observers have a propensity to attend to pseudodiagnostic verbal and non-verbal veracity cues and consequently often exhibit a truth bias [19–22] or at best perform at around chance levels (50: 50) when judging veracity [23–27]. That said, psychological understanding of lay observer judgments is primarily centered on north American and western European English-speaking interactions [28–30]. How lay observers from non-western cultures infer veracity, the cues they attend to, and the decisions they make is not well understood. Yet, culturally defined truth and lie cues do exist [30–33] and so it is sensible to investigate non-western lay-observer behaviours.

Lay-persons truth and lie judgments are the focus of the research reported here, but rather than focusing on cross-cultural interactions, we investigate truth and lie judgments in matched culture, matched native language contexts, where interviews are conducted in Arabic. Twenty-eight nations in western Asia have Arabic as their official language with an estimated 420 million native Arabic speakers globally, yet we found minimal relevant research, and as far as we can ascertain the veracity judgement behaviours of lay observers from this significant cultural group have been neglected.

## Deception and culture

Cultural differences can be defined as 'the collective programming' that distinguish one group from another [34]. Hence, culture typically refers to customary beliefs, values and characteristic societal markers that determine individual attitudes, and shape individual and social behaviours [35–37]. Individuals from different cultures behave and communicate variously in forensic investigative contexts. For example, event details reported can differ across cultures irrespective of veracity, likewise interruptions, assertions of normality and interrupting behaviours which can all be more or less common in both liars and truthtellers as a function of culture [32,33,38,39]. However, all deceivers face a similar challenge, that is how to appear credible. In forensic interview contexts, deceivers may embed lies in the truth and so offer a partially truthful account that includes some deceptive elements, and/or they withhold chunks of information completely, often suggesting a lack of recall or denying having seen or heard elements of the events in question [8,40,41]. Cross cultural understanding of similarities and differences in truth and lie behaviours in interview contexts continues to emerge. However, irrespective of culture, lying is often more cognitively demanding than telling the truth [8,42–46] and so the cognitive challenges of formulating and maintaining a deceptive account in response to investigative questioning appear robust and pancultural.

To appear credible, all deceivers must first formulate a sensible lie and maintain their lie script in response to questions, while holding back the truth to avoid slips of the tongue. Liars must also respond appropriately to follow-up clarification or challenge questions as an interview progresses. Clarification can be in the form of asking for further event details or asking interviewees to repeat answers to monitor consistency. Challenges in an interview typically concern directly questioning whether a response is veridical, or the interviewer might introduce information that has not previously been revealed which casts doubt on the veracity of previous responses. Either way, deceptive interviewees must manage several enduring concurrent cognitive processes [46–49] throughout an interview. In contrast, telling the truth requires the interviewee to recall and verbalise an experienced event, which invokes fewer cognitive resources.

Despite widespread perceptions to the contrary, physical behaviours such as gaze avoidance, self-adaptors, and nervousness have not been found to be reliable cues to veracity because of individual, contextual and cultural variances [50–52]. However, several observable verbal behaviours predicated by cognitive load theories of deception and the impact of attempting to manage several concurrent cognitive processes to appear credible have begun to emerge as potentially stable cues, irrespective of culture and/or language. Deceiver verbal accounts are often impoverished and simple, whereby they provide fewer event specific details than truth-tellers [31,42,50–52]. Although *appearing* to provide informative accounts, in fact deceivers often offer general knowledge information rather than detailed event details that could be verified, for example. Guiding observers to rate detailedness and the amount of verifiable event information has been found to support improved detection of deception performance beyond chance for observers in some circumstances [38].

Truthful accounts are also often more plausible, that is they sound more sensible, likely, or believable. Judgements of plausibility, in terms of how 'believable' an account is, have been found to distinguish truth tellers from liars in some contexts whereby plausibility ratings for deceptive accounts are typically lower than for truthful accounts [30,31,50,51]. Furthermore, observer plausibility ratings have been reported to positively predict the detailedness of an account [40,41]. Lay-observers may, therefore, be able to recognise variable verbal behaviours such as detailedness and plausibility for guiding them in making real-time truth and lie judgments. However, questions arise centred on the real-world challenges for lay-observers.

Cut-off indicators do not exist for the amount of detail an interviewee may or may not provide in response to individual questions or across the interview as a whole, and so subjective assessments are necessary [40,41]. Likewise, plausibility and believability also require subjective assessments concerning how individuals understand how sensible or likely an account is [41]. Subjective assessments are guided by cultural norms, expectations and understandings and so it is important to investigate whether these emerging cues to truth and lies are intersubjective [42]. That is, whether context germane to culture leverages space that might make detailedness and plausibility less important or less valid subjective cues for some cultures.

Second, applied researchers use various paradigms whereby lay-observer participants often view multiple interviews or read multiple interview transcripts where some participants have been incentivised to be deceptive while others are instructed to be truthful [31,40,43,47,53,54]. This approach has been vital for improved understanding of the efficacy of various types of interview approaches for amplifying cues to truth and lies, and how veracity judgments are made. However, in reality lay observers may only have access to a single interview (audio, transcript or video), or one or two relevant portions of an interview which does not allow group performance comparisons. Access to just one interview does not support relative nor comparative verbal behaviour judgements as often occurs in laboratory research paradigms where anomalous verbal behaviour can sometimes be easier to spot across a group, or cohort [55,56]. In such instances, observers can be alerted to truth and lies in a manner that does not map onto the real world where they may be asked to make an absolute judgment based on the behaviours apparent in one interview or transcript.

## The current research

Using first language as a proxy for cultural origins, the research reported here advances understanding of the utility of detailedness, believability and plausibility as veracity cues in forensic interview contexts for lay-observers from a non-western culture, and investigates truth and lie cues attended too. To reflect real-world conditions, observers have access to just one video interview conducted in Arabic, following which they are immediately asked to make a real-time truth or lie judgment. Despite Arabic being one of the most spoken languages worldwide, as far as we can ascertain, lay observer truth and lie judgments of this significant cultural group have received little attention using single culture paradigms. Here, interviewees are mock persons of interest, known to have IRI. Interviews are conducted in Arabic using an information gathering interview forensic protocol. Lay observers all have Arabic as their first language and are asked to self-identify their ethnic/cultural background.

The psychological literature of direct relevance to the to the focus of this research is extremely sparce, and research findings are mixed. However, cultural differences in perceptions of truth and lies behaviours are often reported which can undermine credibility in cross-cultural interview contexts, whereby misunderstanding and misinterpretation typically triggers a lie bias [28,32,33,57–59], although not always [60,61]. To date, it appears researchers have yet to investigate the detection of truth and lies by lay-observers in non-western culture and language matched forensic contexts where observers are unable to make comparative judgments, as is typically the case in the real-world.

The cognitive challenges associated with verbal deception are believed to be pancultural, and so verbal behaviours triggered by cognitive demand, such as levels of event detailedness, believability and plausibility, for example, may transcend culture and language, or they may not. While individuals from Western individualist cultures tend to provide more information than those from non-Western collectivist cultures [62–65], some research has indicated that all truth tellers generally report more details than deceivers, including in Arab and south Asian

cultures [30,31,56,65]. Given the novelty of this research, rather than hypothesising, we formulated a series of research questions towards advancing pancultural understanding with reference to the real-world challenges, current empirical understanding and associated empirical questions raised by those concerned in developing guidance for practice. It is these questions that guided our paradigm and analysis approach, as follows.

First, we examined lay-persons absolute judgments of truth and lies having seen a single within culture interview conducted in matched first language. Consistent with findings from research with other within cultural groups, we expected a truth bias to emerge, and that confidence would differ across veracity conditions.

Second, we examined subjective assessments of plausibility, believability and detailedness as veracity cues, asking whether variations across deceiver and truthteller accounts are discernible and impactful for guiding veracity judgements having been exposed to just one interview. Consistent with a very limited amount of previous research, we expected lay-persons to rate liars as less plausible, less believable and less detailed than truthtellers. We use the terms plausible and believable since although virtually synonymous, statements that are believable are *likely* to be correct or true whereas plausible refers to accounts being *reasonable* in terms of being correct or true. It is unclear whether plausible would be robustly and similarly interpreted across cultural groups, since the term has emerged from Western research paradigms. Research from other domains suggests this may not necessarily be the case [66,67] and so we include believable to cross validate and as a potential additional cue.

Third, we examined the types of cues attended to. We recognise the limited amount of culture specific prior research in this area, nonetheless we expected lay-observers attention would be drawn to behaviours that have previously been found to be pseudo diagnostic.

## Methods and materials

### Ethics

This research was ethically approved by the University of Westminster Research Ethics Committee: ETH1617-0528 and the US Dept of Justice FBI institutional Review Board Docket No. 353–16 and was conducted in accordance with the British Psychological Society and Health and Care Practitioner Council codes of ethical conduct. All participants were adults (18 years and older). In line with internationally recognised ethical best practice, opt-in consent was gained from each lay observer participant via the Qualtrics research platform. Participants first read an information screen explaining information about anonymity, what they would be asked to do and how they were able to withdraw *only* prior to submission of their responses since the study was anonymous, withdrawal post submission would be impossible. Participants were offered the chance to contact the researchers with any additional questions. Participants then moved forward to a consent screen. Participants who did not consent (by clicking the 'I do not consent' button), were not able to take part in the research and so were directed to a generic 'thank you and debrief screen'. Those who consented moved through the research process in Qualtrics as described. Digital records of consent are stored on the Qualtrics platform. The lay-observer data reported in this research was collected between 1st June 2023 and 28th August 2023.

### Forensic interview videos

The two videos used for this research are of interviewees who had been involved in a live event as described below, which was designed to mimic the experiences of some individuals who might later be interviewed as a person of interest. Interviewee performance data and performance has been previously fully reported and the paradigm has also been reported in full and

[8], but for replication purposes and manuscript clarity a brief description of the paradigm is provided.

Culturally matched participants experienced the target event in groups of six. Embedded in each participant group was a culturally matched confederate (C1) playing the role of a participant. Confederate 2 (C2) played the role of researcher (C2). C2 greeted participants and ran the session during which the target event occurred. Participants were primed to be sympathetic to C1 as follows: i) placing C1 within the participant group (C1 arrived and interacted with participants, completed the same tasks, etc.) created conditions for in-group favouritism; ii) the scenario (the manner in which the event unfolded, and verbal exchanges that took place between C1 and C2 during the event); and iii) C2 acting as researcher in charge of the session, creating a perceived imbalance of social influence, in which C2 was portrayed as more powerful than C1.

Both interviewees were male aged 37 (interviewee A) and 38 (interviewee B) years old. Both self-identified as Arabic and Middle Eastern, were bi-lingual and born outside of the UK, with Arabic as their first/native language they had used at home with their parents and family from birth and used at school. English was a second language that interviewees learned from the age of 12 years onwards at school. Both interviewees were paid $30 each to participate and were further incentivised to withhold information (be deceptive) with the offer of an additional payment of $60 dependent upon their interview performance. Of the two interviewee participants taking part in the research reported here, one complied with this instruction (interviewee B), the other did not (interviewee A), thus the interview data comprised one interview with a truthteller one with a deceiver. The interviewee research procedure involved four phases:

1. Interviewees were provided with an information and consent sheet following. Signed consent and audio recorded consent was collected. Following consent, interviewees moved as a group to a seminar room to complete two research questionnaires.

2. While completing the first questionnaire, an unexpected event took place, involving C1 and C2 comprising a verbally aggressive altercation during which a laptop computer was seriously damaged–the entire session, from entering to leaving the event room lasted approximately 25 minutes.

3. Following the event, interviewees were then individually interviewed, and as occurs during real criminal investigations in many jurisdictions (e.g., UK, across the EU and some US States, Australia, and Ireland) and for applied research of this nature, interviews were digitally audio and video recorded. Interviews followed an interview protocol (see Appendix A in S1 Appendix).

## Interviewer

The interviewer was born in the UAE and spoke Arabic as a first language but was fluent in English as a second language. The interviewer underwent a half day of classroom training (given by the first author, an experienced interviewer), which included a detailed explanation of the relevant interview protocol and role-play practice. The interviewer also took part in an additional half-day practice session prior to conducting interviews, which was audio recorded. Detailed verbal feedback was provided on adherence to the protocol. The interviewer was naïve to the design and experimental hypotheses but was provided with the following instructions *'The researcher's computer was seriously damaged during the data collection session. Your job is to interview the people in the room and find out exactly how the damage happened, using the interview protocol'*.

## Interview protocol

Both interviewees were interviewed using the Framed Controlled Cognitive Engagement (Framed—CCE) protocol [3,37]. Interviews comprised five discrete phases as follows: 1) explain and build rapport, 2) free account, 3) probed questioning, 4) challenge, and 5) closure (for protocol see Appendix A in S1 Appendix).

## Lay observer participants

The data collection procedure for lay observers was identical across conditions other than in the cue direction condition where participants were provided with one additional clear instruction prior to viewing the interview, which was to take notice of the amount of detail provided by the interviewee about the event, and to rate the amount of detail on a scale (see materials and Appendix B in S1 Appendix). Further, that the detail rating should then be used to guide their veracity decision. The research was hosted on Qualtrics. Participants were recruited via Prolific platform where participants were screened for age (+ 18 years) and language (reading and speaking Modern Standard Arabic as first language and English as second). Qualifying participants were then directed to Qualtrics via a one-time link where further demographic questions were included (see Appendix B in S1 Appendix).

In both conditions (cue direction and no cue direction) all participants were provided with full information about the study, followed by consent information (in Modern Standard Arabic). All participants then completed a series of demographic questions (age, country of birth, parents first language and schooling). Although we used first language (here Arabic) as a proxy for culture, we also asked participants to indicate their cultural identity in terms of their self-identity and self-perception (see Appendix C in S1 Appendix for self-identity data). Participants were provided with the following written information (in Arabic): '*that the person being interviewed was known to be present during a serious incident where a university laptop was seriously damaged*'. Participants in the cue direction condition were further instructed to '*take note of the degree to which the interviewee includes details such as descriptions of people, places, actions, objects, events, and the timing of events. Think about the degree to which the message of the interviewee seemed complete, concrete, striking, or rich in details*' [35] It was further explained that they would be asked to rate the interviewee's verbal responses to questions for detailedness after they had seen the video, but before making a veracity decision.

All participants then watched the video (either A or B according to veracity condition randomly assigned, conducted in Arabic). Immediately after having viewed the video interview, participants in the cue direction condition were asked to i) make a detailedness judgement ranging from 1 (no detailed) to 10 (very detailed) and ii) were instructed to use the detailedness judgement score to guide their truth or lie dichotomous decision (where 5 and below should trigger a deception decision). Participants in the no cue direction condition did not receive these instructions and so did not make a detailedness judgment. Rather, they simply made a dichotomous truth or lie decision (see Appendix B in S1 Appendix for materials).

All participants completed plausibility and believability scales (all ranging from 1 to 10) and were asked to indicate the types of behaviours (verbal only, behavioural only or a mix of both) that had been important in helping them to make a veracity decision. Participants who had indicted verbal behaviours had been important/partially important were asked to think about the way in which the interviewee had verbally responded to questions and to indicate all that applied from a range of known present verbal behaviours. (see Appendix C in S1 Appendix for materials).

## Stimulus interviews

Each participant listened and watched only one of two information-gathering style interviews. Each interview was conducted in Arabic by the same bilingual interviewer whose first language was Arabic. Both interviews followed the same verbal protocol (see Appendix A in S1 Appendix). In contravention of the confederate (C3) researcher instructions (see procedure), the interviewee in interview A answered all questions truthfully and so did not withhold information nor formulate a deceptive account of the target event. In interview B (deceiver), as instructed by C3 the interviewee was deceptive by withholding elements of the target event, and so provided an account that was deceptive in part (see procedure). Interview A lasted 11.38 minutes. Interview B lasted 11.35 minutes.

The interviews selected for this research are part of a larger, part-published dataset and were selected to control for potentially confounding variances. First all interviews were translated, transcribed and then coded for target event details. We broadly defined details as verbalised event information provided by the interviewee in response to questions, such as (but not limited to) descriptions of the people involved in the incident in question, clear descriptions of the place and time that the incident took place, information regarding the two confederates actions (including speech), the actions of others present (including speech), the objects involved, and the timing of events. Interview A and B were selected because they differed for event detail verbalised by the interviewee whereby interviewee A provided 41 event details, whereas interviewee B provided 17. Hence, there was an 141% difference in the number of event specific details provided across the two interviews.

Second, both interviewees were well matched. Both were Arabic men, aged 33 and 35 respectively and of a similar west Asian appearance with short dark/black hair. Both were clean shaven but with some dark beard regrowth/shadow, and both were wearing casual clothes (t-shirt & shirt) with no visible tattoos, scars nor other distinguishing markings. Both spoke in Arabic from the start to the end of the interview, were interviewed in the same room and at the same position/angle to the interviewer. Both had been born and raised in Libya and had completed their formal schooling in Libya with English as a second language learned after the age of 16 years. Both interviews were of a similar duration (approx. 11 mins 30 seconds each).

## Design and analysis approach

Between subjects 2 Veracity (truth; lie) X 2 Direction (No cue direction; Cue direction) design was employed, collecting dichotomous veracity decision data and scale data for plausibility, confidence and believability from each participant (see procedure and materials). To examine lay-persons absolute judgments of truth and lies we conducted a three-way loglinear analysis of veracity decisions to provide an odds ratio measure of association between an exposure (veracity condition) and outcome (veracity decision). We conducted a series of 2 X 2 ANOVAs to investigate subjective assessments of plausibility, believability and detailedness as veracity cues as a function of veracity condition and cue condition, reporting main effects and interactions. To investigate the relative importance of veracity cues, we conducted Freidman test repeated measures analyses of variance by ranks, followed by Mann-Whitney U across cue conditions (Cue Vs. No Cue).

## Lay observer participants

A total of 217 bilingual participants, fluent in Modern Standard Arabic as a native/first language, took part as lay observers. One hundred and one females (46.5%), 112 (51.6%) males, and one participant who preferred not to say. Mean age was 27.14 years (SD = 7.12), ranging

from 18 to 63 years. While sex and gender are reported, neither are theorized to impact our findings and so sex nor gender-based analyses were not carried out. A-priori power analysis conducted using G*Power 3.1 to determine minimum sample size estimation to detect a practically meaningful medium effect size for applied research of this nature was $N = 179$ (assuming power = 0.80, effect size F 0.25, $a = 0.05$). Thus, the obtained sample size of $N = 217$ was adequate given resource constraints and access to Arabic speaking populations and is in line with sample size norms described in many empirical cross-cultural studies similar to the one reported here [28,31,60,61,68,69].

Observer participants were only able to take part in one condition and were blind to the existence of the additional studies and conditions and were excluded/screened from other conditions via the Prolific targeted screening and exclusion facility. Two hundred participants were recruited from the general population via Prolific and participated during 1st July and 11th August 2023. The remaining 17 were recruited via social media, word of mouth and snowballing, and participated in the first week in August 2023. Mean age as a function of condition veracity X direction condition did not differ significantly, $F(3, 216) = .149$, $p = .930$, likewise gender distribution, $X^2 (3, 216) = 5.049$, $p = .536$.

## Results

### Veracity decisions

Overall, the dichotomous truth/lie decision data revealed a truth bias. Irrespective of condition (truthteller, deceiver, cue direction, no cue direction), 143 (66%) of observers made a truth decision whereas 74 (34%) made a lie decision. Across conditions, counts of correct and incorrect deception and truth decisions are shown in Fig 1 (below).

A three-way loglinear analysis of veracity decisions produced a final model that retained all effects. The likelihood ratio of this model was $\chi^2 (0) = 0$, $p = 1$ indicating the higher order model (veracity condition X cue direction condition X veracity decision) was significant, $\chi^2 (1) = 14.105$, $p < .001$. There was a significant association between veracity condition (truth, deception) and correct veracity decisions, $\chi^2 (1, 217) = 16.471$, $p < .001$. Eighty-six lay-observers (79%) in the truth condition correctly judged the interviewee was truthful, whereas in the deception condition 51(47%) correctly judged the interviewee as deceptive. The odds ratio of making a correct veracity decision were 19.32 times higher in the truth condition than the deception condition.

Chi-square ($\chi^2$) probability distribution revealed a significant association between cue direction (no cue direction, cue direction) and correct veracity decision, $\chi^2 (1, 217) = 11.471$, $p < .001$. Twenty-six lay-observers (23%) in the cue direction condition correctly judged the interviewee as deceptive whereas in the no cue direction condition 48 (43%) correctly judged the interviewee as deceptive. The odds ratio of correctly deciding the interviewee had been deceptive was 2.56 times higher in the no cue direction condition than the cue direction condition. Loglinear analysis indicated lay observers were more likely to make a correct veracity decision when observing a truthteller than a deceiver and were less likely to correctly judge the interviewee as deceptive when provided with cue directions to make a heuristic detailedness judgement to guide their veracity decisions.

### Confidence scale

There was a significant main effect of veracity condition for confidence ratings (on a scale from 1 to 10, where 1 = at all confident and 10 = completely confident), $F(1, 213) = 7.938$, $p = .005$, $\eta_p^2 = .36$. Participants in the lie condition were less confident ($M = 6.60$, SD = 1.93, 95% CI, 6.27, 6.94) than those in the truth condition ($M = 7.28$, SD = 1.60, 95% CI, 6.95, 7.62). The

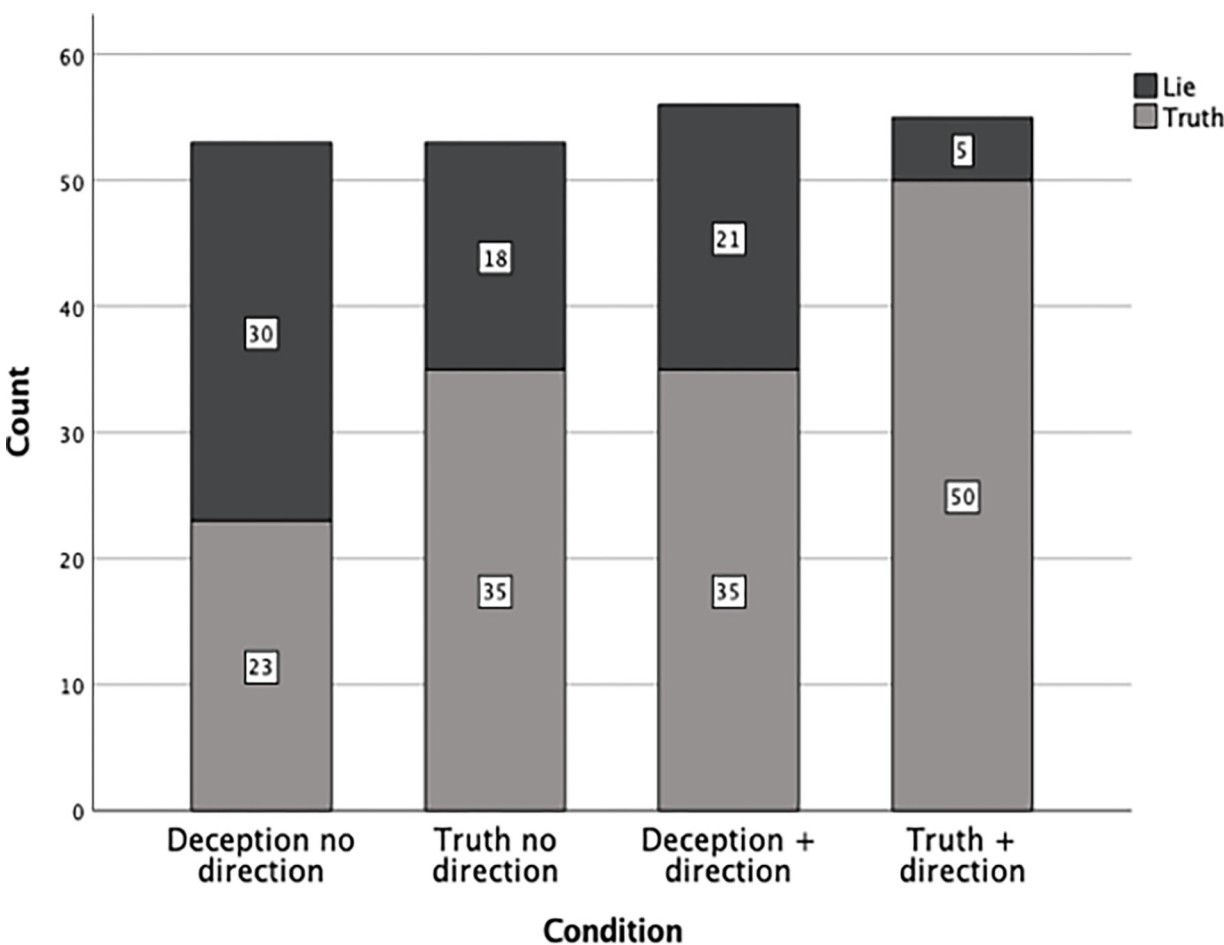

**Fig 1. Veracity decision count as a function of condition** ($N = 217$).

main effect of cue direction ($M_{\text{no direction}} = 6.76$, SD = 1.89, 95% CI, 6.42, 7.10; $M_{\text{cue direction}} = 7.12$, SD = 1.70, 95% CI, 6.79, 7.45) and the veracity X direction interaction were non-significant, all $F$s $< 2.243$, all $p$s $> .136$ (see Table 1 below for interaction means, SDs and 95% CIs).

## Believability scale

The main effects of veracity and cue direction for believability scale ratings (on a scale from 1 to 10, where 1 = not at all believable and 10 = completely believable) were significant, $F(1, 212)$

**Table 1. Mean interaction confidence scale ratings** ($N = 217$).

|  | Mean (SD) 95% CI |
| --- | --- |
| **Condition** |  |
| Truth |  |
| Cue Direction | 7.34 (1.68) 6.87, 7.81 |
| No Cue Direction | 7.23 (1.54) 6.45, 7.71 |
| Deception |  |
| Cue Direction | 6.91 (2.07) 6.44, 7.38 |
| No Cue Direction | 6.30 (1.74) 5.82, 6.78 |

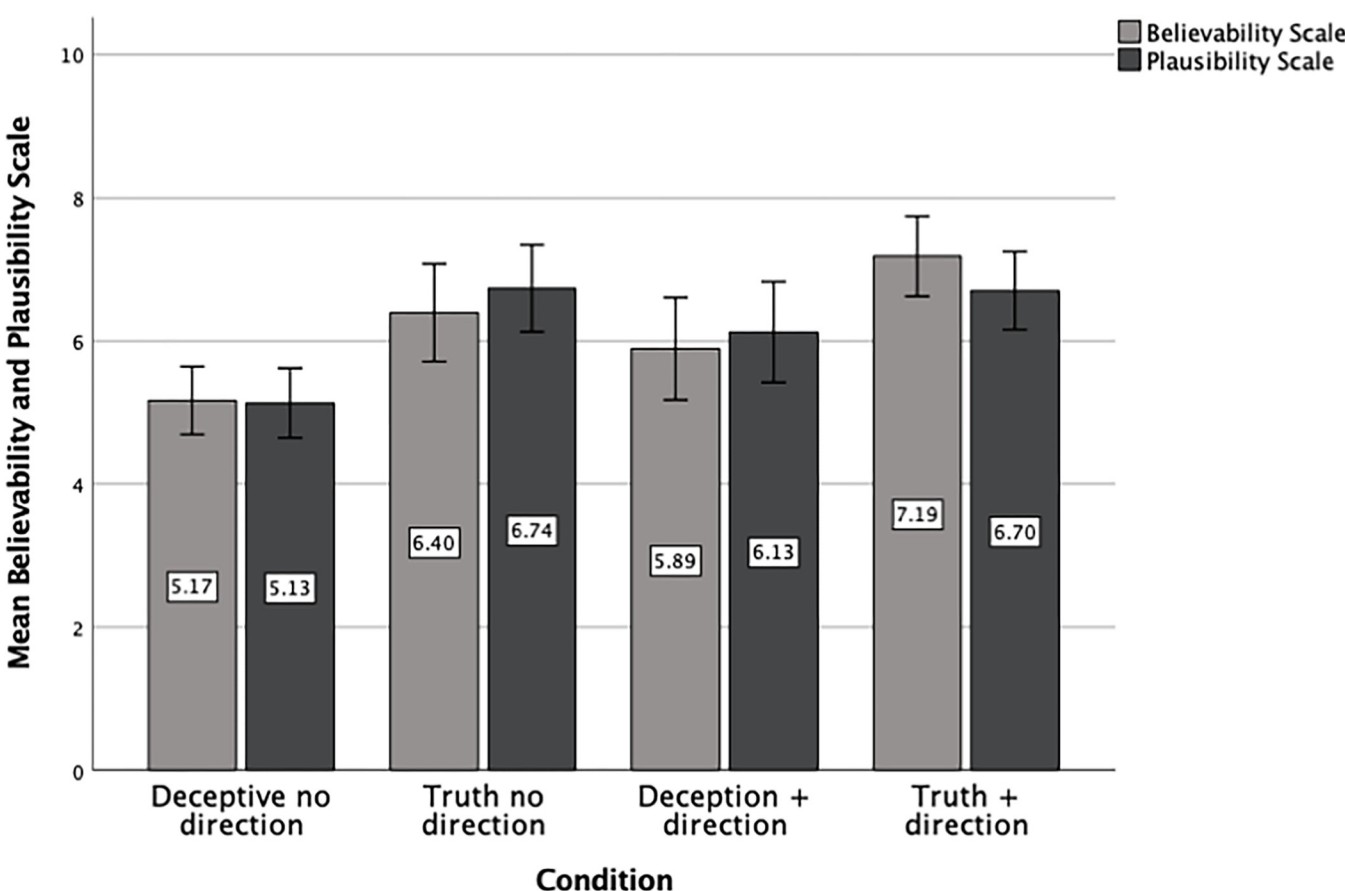

**Fig 2. Believability and plausibility scale interaction means (*N* = 217).**

= 18.701, $p$ < .001, $\eta_p^2$ = .78, and, $F(1, 212)$ = 6.263, $p$ = .013, $\eta_p^2$ = .29, respectively. Participants in the truth condition rated the interviewee as more believable (*M* = 6.83, SD = 2.31, 95% CI, 6.40, 7.26) than those in the deception condition (*M* = 5.50, SD = 2.26, 95% CI, 5.07, 5.93) and participants in the cue direction condition rated the interviewee as more believable (*M* = 6.55, SD = 2.48, 95% CI, 6.13, 6.97) than those in the no direction condition (*M* = 5.78, SD = 2.21, 95% CI, 5.35, 6.22). The veracity X cue direction interaction was non-significant, F = .062 $p$ = .804 (see Fig 2 below for interaction means).

### Plausibility scale

The main effect of veracity for plausibility scale ratings (on a scale from 1 to 10, where 1 = not at all plausible and 10 = completely plausible) was significant, $F(1, 212)$ = 15.257, $p$ < .001, $\eta_p^2$ = .67. Participants in the truth condition rated the interviewee as more plausible (*M* = 6.75, SD = 2.10, 95% CI, 6.34, 7.16) than those in the deception condition (*M* = 5.59, SD = 2.27, 95% CI, 5.18, 6.01). The main effect of cue direction (*M* <sub>no direction</sub> = 5.93, SD = 2.15, 95% CI, 5.51, 6.35; *M* <sub>direction</sub> = 6.41, SD = 2.35, 95% CI, 6.00, 6.82) and the veracity X cue direction interaction were both non-significant, all Fs < 2.575, all *p*s > .110 (see Fig 2 below for interaction means).

A repeated measures ANOVA comparing plausibility and believability ratings across conditions revealed non-significant main effects and a non-significant interaction, all *F*s < 5.790, all *p*s, > .017 across conditions.

## Detailedness

Only participants in the cue direction condition completed a detailedness scale. Analysis as a function of veracity decision revealed participants in the truth condition rated the interviewee's account as significantly more detailed ($M = 7.07$, SD = 1.46, 95% CI 6.63, 7.46) than those in the deception condition ($M = 5.96$, SD = 1.78, 95% CI 5.43, 6.39), $F(1, 109) = 14.078$, $p < .001$, $\eta_p^2 = .11$.

## Veracity cue scale

Analysis of the cues lay observers indicated were important when making a veracity decision revealed significant main effects of veracity and cue direction for cue scale ratings (1 = verbal only, 2 = mainly verbal but some behaviour, 3 = verbal and behaviour equally, 4 = mainly behaviour but some verbal, 5 = only behaviour), $F(1, 213) = 18.775$, $p < .001$, $\eta_p^2 = .81$, and, $F(1, 213) = 17.705$, $p < .001$, $\eta_p^2 = .92$, respectively. Irrespective of cue direction condition, participants in the truth condition ($M = 2.79$, SD = .97, 95% CI, 2.62, 2.96) indicated that they had *mainly* listened to what the interviewee was saying but took some notice of how the interviewee behaved. Conversely, participants in the lie condition had *mainly* taken notice of how the interviewee behaved but took some notice of what the interviewee said ($M = 3.32$, SD = .91, 95% CI, 3.15, 3.49).

In a similar vein, irrespective of veracity condition, participants in the cue direction condition (who had been cued to note how much detail the interviewee provided and to rate detailedness) indicated they had *mainly* listened to what the interviewee was saying but also took some notice of how the interviewee behaved when making a veracity decision ($M_{\text{cue direction}} = 2.77$, SD = .82, 95% CI, 2.60, 2.94 –see Table 2 below for interaction means).

Participants in the no cue direction condition *mainly* took notice of how the interviewee behaved but took some notice of what the interviewee said ($M_{\text{no direction}} = 3.34$, SD = 1.03, 95% CI, 3.16, 3.51). The cue condition X veracity interaction was non-significant $F = .001$, $p = .974$.

## Verbal behaviours

Overall, of the 217 lay-observers, 156 (72%) responded that they had primarily or equally attended to verbal behaviours (by responding 1, 2 or 3 on the veracity cue) of which just 10 (4%) responded they had *only* attended to verbal behaviour, 50 (23%) that they had *mainly* listened to what the interviewee said but had taken *some* notice of how the interviewee behaved, while 96 (44%) stated they had attended equally to what the interviewee said and how they behaved.

Participants indicated which verbal behaviours had been influential when making veracity decision (they were able to indicate more than one behaviour). A Freidman test to rank the

**Table 2.  Cue scale interaction means, SDs and 95% CIs ($N = 217$).**

|  | Mean (SD) 95% CI |
| --- | --- |
| **Condition** |  |
| Truth |  |
| Cue Direction | 2.50 (.85) 2.26, 2.74 |
| No Cue Direction | 3.08 (1.01) 2.83, 3,32 |
| Deception |  |
| Cue Direction | 3.04 (.75) 2.79, 3.28 |
| No Cue Direction | 3.60 (.99) 3.36, 3.85 |

**Table 3. Mean ranking, numbers of responses and percentages for verbal behaviours ($n$ = 156).**

| Verbal Behaviour | Mean Rank | Number (%) |
|---|---|---|
| Variously correcting answers | 3.94 | 94 (60.25) |
| Giving a consistent account | 3.55 | 74 (47.43) |
| Providing a lot of detail | 3.52 | 72 (46.15) |
| Giving a simple account | 3.44 | 68 (43.59) |
| Talking about other people present | 3.36 | 64 (41.02) |
| Giving a complicated account | 3.19 | 55 (35.25) |

verbal behaviours attended to was significant, $\chi^2$ (156) = 21.232, $p$ < .001 (see Table 3 below). Overall, self-corrections were ranked as most influential and providing a complicated account was ranked as least influential.

Further Freidman tests to rank the verbal behaviours attended as a function of cue direction condition was significant for the no cue direction condition, $\chi^2$ (46) = 20.181, $p$ < .001, (see Table 4) but non-significant for the cue direction group, $\chi^2$ (110) = 9.032, $p$ = .108 (see Table 4). Again, self-corrections were ranked as most influential for the non-cue direction group and providing a complicated account was ranked least influential.

Mann-Whitney tests revealed significant differences in verbal behaviours attended too as a function of cue direction condition (direction; no direction) for consistency, U = 1920.00 $p$ = .006 (Mean Rank $_{\text{No direction}}$ = 65.24; Mean Rank $_{\text{Cue direction}}$ = 84.05), complexity, U = 1967, $p$ = .008 (Mean Rank $_{\text{No direction}}$ = 66.26; Mean Rank $_{\text{Cue direction}}$ = 83.62), and detailedness, U = 2044.00, $p$ = .029 (Mean Rank $_{\text{No direction}}$ = 67.93; Mean Rank $_{\text{direction}}$ = 82.94). Participants in the cue direction group attended to all three verbal behaviours more than those in the no cue direction. Mean rankings for all other verbal behaviours were non-significant across the two groups, all Us > 2136.00 $p$s, > .075, indicating participants attended equally across conditions.

## Results summary

The truthful interviewee was accurately detected by 79% of observers whereas accuracy dropped to just below chance levels (47%) in the deceptive interviewee condition. Overall, the truthful interviewee was rated as providing a more detailed account than the deceptive interview. However, giving specific instructions to look for the detailed responses for cueing truth and lies significantly undermined observer ability to accurately detect the deceptive interviewee, falling to 23% accuracy. Comparatively, detection of deception performance improved in the absence of cue instructions, but again performance was still below chance at 43% accuracy.

**Table 4. Mean ranking for verbal behaviours as a function of no cue ($n$ = 46) and cue direction condition ($n$ = 110).**

| No Cue Direction Mean Rank | | Cue Direction Mean Rank | |
|---|---|---|---|
| Variously correcting answers | 4.33 | Variously correcting answers | 3.78 |
| Talking about other people present | 3.48 | Giving a consistent account | 3.64 |
| Providing a lot of detail | 3.41 | Providing a lot of detail | 3.56 |
| Giving a simple account | 3.41 | Giving a simple account | 3.45 |
| Giving a consistent account | 3.35 | Talking about other people present | 3.31 |
| Giving a complicated account | 3.02 | Giving a complicated account | 3.26 |

Confidence ratings were higher in the truthful interviewee condition and confidence remained stable irrespective of detailedness cue instructions. Overall, the truthful interviewee was rated as more believable and more plausible than the deceptive interviewee. Detailedness cueing did impact believability ratings whereby lay observers in the cue condition rated the interviewee as more believable.

Participants viewing the truthful interviewee stated that they mainly took account of verbal behaviour when making their veracity decision but took some notice of nonverbal behaviours. Conversely, those viewing the deceptive interviewee mainly took note of nonverbal behaviour but took some account of what the interviewee said. This pattern of results was mirrored across the cue conditions whereby participants in the cued condition mainly took account of verbal behaviour but took some notice of nonverbal behaviour while those in the no cue condition mainly noted nonverbal behaviour but took some note of verbal behaviours.

Overall, the verbal behaviour used by all lay observers to cue their veracity decision was 'correcting answers' whereas the least important was 'giving a complicated account'. The pattern of cues attended too across all conditions was similar.

## Discussion

Behavioural and verbal norms and expectations vary across cultures and so research understanding truth and lie judgements in non-western cultures plays a central role in advancing knowledge and reducing misunderstanding and misinterpretation. Towards improved ecological validity we focused on the lay observer (non-student) veracity judgements of participants with Arabic as their native language who variously self-identified as Arabic, Middle Eastern, North African and/or West Asian etc. Furthermore, to improve generalizability we mirrored common real-world practice whereby our lay observers had access to just a single interview, following which they were asked to make an absolute veracity judgement.

Consistent with findings from research carried out with North American and Western European participants, and as predicted by Truth Default Theory [70], overall, our results reveal a truth bias whereby 66% of lay participants made a truth judgment. Most interpersonal communication in everyday life is truthful and assuming truthfulness appears pan cultural for within culture non-western native speaker interactions, as has been widely reported by others, albeit typically where relative veracity judgments have been made [22,53,59,68,71,72]. Judgements as a function of veracity condition revealed participants in the truth condition were most accurate whereby 79% correctly judged the interviewee as truthful. Accuracy was just below chance at just 47% in the deception condition. Again, this pattern of results mirrors the findings of research with North American and Western European observer participants and some non-western native speaker research [48,73,74], although little research has investigated truth and lie judgements following a single interaction.

Turning to the impact of instructing observers to take note of verbal detailedness and then to subsequently use these ratings as a rule of thumb to cue a veracity decision. Our results deviate from the findings of others [40] where detailedness was found to be an important single heuristic for guiding accurate judgments. Here, relatively our lay observers performed better when judging deception in the absence of a detailedness cue direction (albeit at just below chance). Accuracy dropped considerably to just 23% when instructed to use the detailedness cue. Although our findings reveal significant differences in detailedness ratings across the veracity conditions, the mean detailedness rating in the deception condition was 5.96, which is very near to the truth cut-off of six. Cautious dichotomous decisions in terms of erring on the side of truth as often occurs in situations of uncertainty may account for this result, but this may also be a cultural effect.

Different cultures vary in their linguistic communication style [75,76] While cautious dichotomous decisions may have been triggered by the nature of the cut-off instructions, cultural norms may also have played a part in why detailedness may not have been a diagnostic cue here. Participants in the no-cue condition indicted taking note of detailedness even though they were not directed towards any verbal behaviours and so detailedness appears relevant to some degree. However, repetition, re-phrasing and reverse re-phrasing are common persuasive strategies for Arabic speakers, none of which necessarily introduce additional detail across the duration of an interview. This cultural norm may have interfered with participant understanding of the notion of detail and what more or less detail 'looks' like in practice.

Both cue conditions did rank 'corrections' as being the most noted verbal behaviour despite instructions indicting otherwise in the cue direction condition. Corrections may be interpreted as a red flag for Arabic speakers, whereby language is viewed as a 'container' of truth and knowledge and repetitions are common and to be expected. Conversely, alterations and corrections may be uncomfortable and seen as anomalous. It has been suggested that individuals from high-context Asian and Arabic cultures would not naturally expect to be specific when communicating. They may be slower making a point when answering questions than individuals from low context North America and Western Europe cultures. This preference for what is often referred to as roundabout messaging can be even more marked if the topic or 'message' is unpleasant or difficult [69,77,78] as was the case here.

The witnessed scenario depicted a verbally aggressive exchange and damage to a university laptop in conditions of perceived imbalance of social influence. Despite instructions to provide as much detail as possible, answering questions may have felt unpleasant and/or difficult and so one might expect verbal accounts to be less detailed *per se*, irrespective of veracity condition. The liar account used for this research was far less detailed than the truthteller account in terms of the number of event details provided. Directing participants to consider detailedness may have inadvertently alerted observers in the cue condition to note a culturally determined truth cue: less detail being a cultural norm and to be expected particularly when truthfully answering questions about an unpleasant incident.

Where comparative judgements are not possible, our results indicate a lack of detail may be a less valid veracity cue for some non-western cultural groups. Comparative judgements allow observers to naturally notice anomalous or seemingly 'different' behaviours across a cohort, which can then be used to guide truth or lie decisions. Differences in levels of detailedness across a within-culture group might be an effective cue in repeated measures contexts, but we did not find this to be a useful heuristic. Others have reported that individuals from some cultures (e.g., Arab and Chinese) naturally provide less detail than individuals from western cultures and so perceptions regarding a lack of detail may not emerge as a pancultural red flag, although more research is needed [10,31,23]. Interaction effects highlight observers rated detailedness differently across cue conditions, but these differences did not impact veracity judgments, and detailedness was not ranked as the most important verbal cue behaviour irrespective of cue direction condition.

Ratings of believability and plausibility were used to further understand how observers made their judgements. Although virtually synonymous, given the non-western nature of this research we decided to use both terms for cross validation purposes since plausibility has emerged from North American and Western European research. Main effects for both subjective ratings were consistent with previous research and add to suggestions that plausibility is a cue worth examining [51]. Participants all rated the truthteller as more plausible and more believable than the liar. However, participants in the cue direction condition (who were directed to attend to detailedness) rated both truthteller and liar as more believable than those

in the no cue direction condition, indicating caution when considering subjective measures of believability combined with cueing detailedness.

Previous research has found differences in the believability of individual liars can explain 98% of variance in accuracy of deception detection [59]. It is possible that despite our attempts to match the two interviews, the deceptive interviewee may have had an honest demeanour [79], albeit this is speculative. Given the attributes of demeanour are complex to specify and were neither considered nor controlled, this finding may be a materials effect. It appears therefore, that plausibility and believability have potential as robust pan cultural subjective assessments, but believability may be more linked to behavioural demeanours which were not the focus of this research.

Our results also reveal that participants in the cue condition did as instructed and had *mainly listened to what the interviewee was* saying but took some notice of how the interviewee behaved when judging veracity. Conversely, participants in the no cue direction condition indicated they *mainly took notice of how the interviewee behaved* but took some notice of what the interviewee said. This latter finding suggests non-verbal indicators may have been effective for correctly identifying truth since overall participants in the no-cue direction condition were more likely to make a correct truth judgment.

Non-verbal gestures are an important aspect of communication in many non-western high context cultures which may explain the importance of this cue for our observers. However, perceptions do not necessarily mirror behaviour, and given the nature of our paradigm there is little reason to question widely reported findings that non-verbal cues are not reliable cues to truth and lies [25,34,80–82] even when participants are trained in identifying them [51,83]. Most participants took note of non-verbal behaviours to a lesser or greater degree. It is impossible to tease apart the relative contribution of verbal and non-verbal cues here, thus further research investigating non-western, within-culture non-verbal behaviours is needed to shed more light on this finding.

## Limitations

Better understanding of human behaviours across a broader range of within culture interactions is needed towards diluting western dominated psychological understanding of truth and lies. Furthermore, researchers must consider ecological validity [84]. While our research is novel and relevant to both agendas, as with all experimental research of this nature there are several limitations. Methodological limitations stem from the artificial nature of our judgment task whereby participants were provided with limited detail about the event in question, only viewed one video interview, and their veracity decisions had no ramifications. Future research might consider emphasising the potential implications of accurate versus inaccurate veracity decisions by incentivising participant decision-making perhaps. Allowing participants to view a series of interviews, employing think aloud methods would also help to pick apart the cues noted by this under researched cultural group, particularly the relevance of corrections for example.

Numerous questions emerge centred on the importance or otherwise of corrections, repetitions and detailedness. We did not analyse these linguistic elements in any detail across our larger data set to understand whether the two interviews selected for this research were representative of our Arabic interviews. However, guided by the results of this study, we intend to pursue this aspect of information gathering interviews with Arabic speakers. Finally, we employed an information gathering protocol that includes best practice techniques for gathering IRI and has been used pan culturally in the field [26,37,60,73,85]. However, for this cultural group alternative interview approaches may prove more effective for leveraging potentially diagnostic veracity signals in forensic contexts.

## Conclusions

Behavioural and verbal norms and expectations vary across cultures, however our findings suggest irrespective of culture, lay observers may exhibit similar truth biases and with comparable levels of confidence. Participants in the truth condition were more confident and more accurate, again indicating cues to truthfulness seem more apparent to observers albeit they may not necessarily be quantifiable, which has been reported by others [43,86]. Arabic speaking observers exhibited some similarities but also some notable differences to Western and North American lay observers in the verbal cues they attended too. This finding indicates professional practice must move to recognize that, irrespective of similar cognitive demands associated with truth and lies, linguistic veracity cues appear culturally specific in terms of veracity cue availability and cues being attended too, despite clear instructions to the contrary. Finally, a one size fits all approach to gathering information during forensic interviews may be inappropriate for leveraging culturally salient verbal veracity cues and so researchers must seek to consider how to amplify veracity cues across cultures.

## Supporting information

**S1 Appendix.**
(DOCX)

## Acknowledgments

We would like to thank Amelia Dickinson for her work as a part-time Research Assistant, data manager, and coder on this project.

## Author Contributions

**Conceptualization:** Coral J. Dando, Alexandra L. Sandham, Paul J. Taylor.

**Data curation:** Coral J. Dando, Alexandra L. Sandham.

**Formal analysis:** Coral J. Dando, Paul J. Taylor.

**Funding acquisition:** Coral J. Dando.

**Methodology:** Alexandra L. Sandham, Charlotte Sibbons, Paul J. Taylor.

**Project administration:** Alexandra L. Sandham, Charlotte Sibbons.

**Resources:** Coral J. Dando.

**Writing – original draft:** Coral J. Dando, Alexandra L. Sandham, Charlotte Sibbons.

**Writing – review & editing:** Coral J. Dando, Alexandra L. Sandham, Charlotte Sibbons, Paul J. Taylor.

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
