## [Decision Letter · Decision Letter 0]

30 Jul 2024

PONE-D-24-12342Arabic within culture investigative interviews: Arabic native speaking lay-observer truth and lie accuracy, confidence, and verbal cue selection.PLOS ONE

Dear Dr. Dando,

Thank you for submitting your manuscript to PLOS ONE. After careful consideration, we feel that it has merit but does not fully meet PLOS ONE’s publication criteria as it currently stands. Therefore, we invite you to submit a revised version of the manuscript that addresses the points raised during the review process.

We look forward to receiving your revised manuscript.

Kind regards,

Dipima Buragohain

Guest Editor

PLOS ONE

“This work was funded by the U.S. Government’s High-Value Detainee Interrogation Group (HIG) awarded to the first author - Contract DJF-16-1200-V-0000737. Statements of fact, opinion and analysis in the article are those of the authors and do not reflect the official policy or position of the HIG or the U.S. Government.

We would like to thank Amelia Dickinson for her work as a part-time Research Assistant, data manager, and coder on this project.”

“This work was funded by the U.S. Government’s High-Value Detainee Interrogation Group (HIG) awarded to the first author - Contract DJF-16-1200-V-0000737. Statements of fact, opinion and analysis in the article are those of the authors and do not reflect the official policy or position of the HIG or the U.S. Government.”

4. Please remove your figures from within your manuscript file, leaving only the individual TIFF/EPS image files, uploaded separately. These will be automatically included in the reviewers’ PDF.

Additional Editor Comments:

Dear Author,

Thank you for your paper. Kindly revise the manuscript as per the reviewer feedback and resubmit. Thank you.

Reviewers' comments:

Reviewer's Responses to Questions

**Comments to the Author**

1. Is the manuscript technically sound, and do the data support the conclusions?

Reviewer #1: No

Reviewer #2: Yes

2. Has the statistical analysis been performed appropriately and rigorously? 

Reviewer #1: Yes

Reviewer #2: Yes

3. Have the authors made all data underlying the findings in their manuscript fully available?

Reviewer #1: Yes

Reviewer #2: No

4. Is the manuscript presented in an intelligible fashion and written in standard English?

Reviewer #1: Yes

Reviewer #2: Yes

5. Review Comments to the Author

Reviewer #1: I think that the topic is interesting and the discussion is good. There are some problems in the language and I believe that the author should review the paper. Also, I think that the author should be clear in presenting his ideas .

Reviewer #2: Strengthen the Introduction: While the introduction provides a solid background, it could benefit from a clearer articulation of the research gap and the specific contributions of the study. A more direct statement of the study's objectives would enhance clarity.

Enhance the Discussion Section: The discussion could be expanded to more thoroughly explore the implications of the findings in relation to existing literature. Comparing the results with previous studies on deception detection could provide deeper insights into the significance of the findings.

Clarify Statistical Analysis: While the results are presented clearly, providing more detail on the statistical methods used for analysis would enhance transparency. Including information on the software used and the rationale for selecting specific tests could improve the rigor of the methodology.

Visual Aids: Incorporating more visual aids, such as graphs or tables, to summarize key findings would enhance the presentation of results and make the data more accessible to readers.

Conclude with a Stronger Summary: The conclusion could be more impactful by summarizing the key findings and implications in a concise manner. A strong concluding statement that encapsulates the overall significance of the research would leave a lasting impression on readers.

6. PLOS authors have the option to publish the peer review history of their article (what does this mean?). If published, this will include your full peer review and any attached files.

Reviewer #1: **Yes: **Yasir Alotaibi

Reviewer #2: **Yes: **Atika Dyah Perwita

---

## [Author Response · Author response to Decision Letter 0]

12 Aug 2024

Dear Editor & Reviewers

We thank the reviewers for their comments, which are very similar, largely stylistic in nature, and rather general in terms of perceived manuscript conventions. We have addressed each of the comments as appropriate below, and in the resubmission. We have addressed all editorial requirements (see below). Finally we have addressed the in-text comments added to the manuscript by the reviewer and adjusted the manuscript where appropriate or responded where adjustments were not/could not be made. 

We look forward to hearing from you in due course.

Regards

Prof Coral Dando PhD.

Editorial Comments

1. We have removed funding related statements form the acknowledgements.

2. We have removed figures from within the manuscript and uploaded these separately as directed.

3. There are no alterations to refences list other than adding references relevant to the reviewer comments. 

4. We have adjusted the data viability statement to make our raw data available on acceptance.

Reviewer 1

1. “Introduction” must state the research gap, prior studies on the subject (including those that differ from the prior studies), and the relationship of research problems to the broader interest.

In our initial submission, we had made the research gap very clear throughout the introduction, and the research itself is clearly situated in the ‘real world’ and as such links the research to a broader challenge for national and international criminal justice. For example, paragraphs 3 and 4 of the Introduction clearly highlight the importance of this research to the broader interest. The ‘Deception and Culture’ section cites numerous studies of relevance but also highlights how these studies to not meet the clear gap in understanding as highlighted throughout this section. The ‘Current Research’ section moves to precis all the preceding information to funnel down to offer a clear rational. We are able to offer additional examples if requested. 

2. “The Current Research” must state the focus of research that is different from the previous research

In the second paragraph of the ‘The Current Research’ section of the introduction we have more clearly stated how this research differs from the previous research – specifically that there is no research of direct relevance in terms of mimicking the real world, but that some loosely related research does exist but that this provides mixed findings which may loosely ‘speak’ to elements of the problem space outlined in this manuscript, but does not offer strong insight. We have referenced the published research throughout the Introduction to allow readers to access the primary sources (which is best practice), thus allowing us to keep an acceptable world count for this manuscript towards ‘presenting and discussing our research concisely’ as required for publication in PLOSONE.

3. “Discussion”: must state the research findings, namely the research's keywords that might serve as a guide for problem-solving, based on literature reviews/theoretical framework.

We do not really understand this comment. The research findings in terms of the primary results and their implications have all been discussed throughout the Discussion in turn. But, towards trying to resolve this reviewer’s comment and so as not to slow the review process still further we i) believe problem solving frameworks are not relevant for this manuscript, particularly given the research paradigm and research questions posed in the itroduction, ii) we have already refenced numerous highly relevant, current theoretical frameworks and associated research publications both in the Introduction and Discussion, using each to allow us to interpret our findings – e.g., truth default theory, cognitive load theory, linguistic communication style. Our results are clearly linked to all of these theorical frameworks, throughout the discussion, and discussed in this regard, in turn, iii) simply re-stating of the results at the start of the discussion would be far too repetitive in our opinion, particularly given the comments/requirements of reviewers provide a summary of the results again at the end of the results section – which we have now done – hence a third repetition of the results would be unnecessary and would be in contravention of the PLOSONE publication requirements in terms of ‘presenting and discussing research concisely’. 

Reviewer 2

1. Abstract must present the research problem, method, main findings, conclusions and implications of the main findings.

It is our understanding that PlosOne do not encourage nor request abstracts to be written with subheadings, which may have resulted in a lack of clarity for this reviewer. We had already included the research problem, main findings, conclusion and implications elements in the original submission, but the implications element may not have been as strong as it could be and so we have strengthened this as requested by this reviewer. We have also revisited the abstract in its entirety to improve clarity for this reviewer in terms of strengthening all elements while ensuring the abstract is succinct, while providing a fair overview.

2. “Introduction” must state the research gap, prior studies on the subject (including those that differ from the prior studies), and the relationship of research problems to the broader interest.

This is an exact replication of a comment from reviewer 1 – please refer to our response to his/her comments above.

3. “The Current Research” must state the focus of research that is different from the previous research

This is an exact replication of a comment from reviewer 1 – please refer to our response to his/her comments above.

4. “Discussion”: must state the research findings, namely the research's keywords that might serve as a guide for problem-solving, based on literature reviews/theoretical framework.

This is an exact replication of a comment from reviewer 1 – please refer to our response to his/her comments above.

5. In order for non-experts in the subject of study as well as experts to grasp, figures 1 and 2, as well as tables 1-4, must be presented in a clear and understandable manner.

PLOSONE is a broad audience publication that publishes primary research that contributes to the base of scientific knowledge. Thus, it is our understanding that PLOSONE is not a practitioner outlet and as far as we are aware not aimed at non-experts in terms of scientific rigour, data analysis nor results reporting. Our tables and figures are APA/PLOSONE formatting compliant and include all the required statistical information to allow replication as well as quick and easy access to the relevant results in more detail, thus supplementing the written information. Were we to reduce the number of tables, the results section would become far more difficult to navigate since we would have to either i) produce a very large table with all of the results (means, SDs, 95% CIs) for all of the analyses combined which would be difficult to navigate for all readers irrespective of experience OR ii) include all of this information within the text which would result in a very text heavy results section, thus making this element rather turgid and again very difficult to navigate for all readers. Our Figures are clear and APA complaint, and understandable for all, irrespective of experience and expertise. As such we retain the tables and figures in their current from, but would be happy to discuss further following editorial guidance. 

6. "Conclusion" must present the main findings in accordance with the research objectives, the implications of the main research findings in a broader context, and research directions that can be carried out in the future.

We had already provided a conclusion in the original submission but have now strengthened this element of the manuscript to support reviewer understanding, but are only able to offer a cautious and scientifically robust conclusion, and will always retain this stance, since this is just one study. We had already offered suggestions for future research, and so retain this information. 

In-text manuscript comments

One of the pdf files simply has comments on the manuscript that appear to act as an aid memoir for the reviewer – this type of review/commentary is unusual and having looked at the commentary, all points have are already covered in the initial manuscript submission.

In the second file a series of interesting comments have been offered many of which have been addressed.

1. Additional references re. culture have been added earlier in the introduction and the Refs. Section adjusted accordingly. 

2. The ‘problem space’ has now been defined earlier in the Introduction.

3. We have included ‘analysis approach’ information to aid understanding.

4. We have added a brief results summary at the end of the results section –we are not entirely comfortable with this repetition, the reviewers believe this is necessary and so we have complied

5. All of the results have been very clearly linked to the existing literature (such as there is) in the discussion in the initial submission as one would expect. We have revisited this element of the manuscript but given the paucity of directly relevant research we retain the format of the original discussion in this regard and would not be comfortable in overstating nor over discussing. 

6. Again, we have already clearly interpreted the results in the discussion and linked all back to the relevant literature in the initial submission. 

7. We have added a stronger concluding statement BUT we retain our cautious stance as guided by best practice whereby researchers should not overreach in terms of significance of their research and application implications – to that end we can only offer a cautious and scientifically robust conclusion, and will always retain this stance, since this is just one study.

8. Videoing of interviews is not innovative – this is typical research practice in interviewing contexts (to allow analysis) and gold standard investigative practice in many jurisdictions, worldwide – we have adjusted the manuscript method to explain this to readers who may be unfamiliar with/unaware of this approach. 

9. We do not include a research process diagramme for the interviewee method, since this element of the research paradigm has been summarised in this manuscript only for clarity and for replication purposes. This element is not part of the method for the research reported here, and as we have highlighted is fully published elsewhere, and data is available on OSF. However, we concede that by including so much detail about the interviewees in the first submission we have given the false impression that these data are part of this research. Accordingly, we have adjusted the manuscript and referred to the interviews as forensic interview videos, and reduced the procedural information in the methods section, but we have included a reference as to where this information can be found in full. We are sorry for this confusion and hope this adjustment improves clarity. 

10. Ethical comments have been addressed, but we are surprised at having to include so much additional fine grain detail in a manuscript of this nature, but hopefully this will meet reviewer requirements.

---

## [Editor Report · Decision Letter 1]

30 Aug 2024

Arabic within culture forensic interviews: Arabic native speaking lay-observer truth and lie accuracy, confidence, and verbal cue selection.

PONE-D-24-12342R1

Dear Dr. Dando,

We’re pleased to inform you that your manuscript has been judged scientifically suitable for publication and will be formally accepted for publication once it meets all outstanding technical requirements.

Kind regards,

Dipima Buragohain

Guest Editor

PLOS ONE
---

## [Editor Report · Acceptance letter]

13 Sep 2024

PONE-D-24-12342R1 

PLOS ONE

Dear Dr. Dando, 

I'm pleased to inform you that your manuscript has been deemed suitable for publication in PLOS ONE. Congratulations! Your manuscript is now being handed over to our production team.

Kind regards, 

on behalf of

Dr. Dipima Buragohain 

Guest Editor

PLOS ONE